# Nuclear Receptor Binding Protein 2 Is Downregulated in Medulloblastoma, and Reduces Tumor Cell Survival upon Overexpression

**DOI:** 10.3390/cancers12061483

**Published:** 2020-06-06

**Authors:** Anqi Xiong, Ananya Roy, Argyris Spyrou, Holger Weishaupt, Voichita D. Marinescu, Tommie Olofsson, Ola Hermanson, Fredrik J. Swartling, Karin Forsberg-Nilsson

**Affiliations:** 1Department of Immunology, Genetics and Pathology, Science for Life Laboratory, Uppsala University, 751 85 Uppsala, Sweden; anqi.xiong@ki.se (A.X.); Ananya.roy@igp.uu.se (A.R.); argyris@kth.se (A.S.); holger.weishaupt@igp.uu.se (H.W.); fredrik.swartling@igp.uu.se (F.J.S.); 2Department of Medical Biochemistry and Biophysics, Karolinska Institutet, 171 77 Stockholm, Sweden; 3Department of Intelligent Systems, KTH Royal Institutes of Technology, 100 44 Stockholm, Sweden; 4Department of Medical Biochemistry and Microbiology, Science for Life Laboratory, Uppsala University, 751 23 Uppsala, Sweden; voichita.marinescu@imbim.uu.se; 5National Board of Forensic Medicine, 752 37 Uppsala, Sweden; biodocent@hotmail.com; 6Department of Neuroscience, Karolinska Institutet, 171 77 Stockholm, Sweden; ola.hermanson@ki.se

**Keywords:** brain tumor, pseudokinase, NRBP, pediatric cancer, apoptosis

## Abstract

Pseudokinases, comprising 10% of the human kinome, are emerging as regulators of canonical kinases and their functions are starting to be defined. We previously identified the pseudokinase Nuclear Receptor Binding Protein 2 (NRBP2) in a screen for genes regulated during neural differentiation. During mouse brain development, *NRBP2* is expressed in the cerebellum, and in the adult brain, mainly confined to specific neuronal populations. To study the role of NRBP2 in brain tumors, we stained a brain tumor tissue array for NRPB2, and find its expression to be low, or absent, in a majority of the tumors. This includes medulloblastoma (MB), a pediatric tumor of the cerebellum. Using database mining of published MB data sets, we also find that NRBP2 is expressed at a lower level in MB than in the normal cerebellum. Recent studies indicate that MB exhibits frequent epigenetic alternations and we therefore treated MB cell lines with drugs inhibiting DNA methylation or histone deacetylation, which leads to an upregulation of NRBP2 mRNA expression, showing that it is under epigenetic regulation in cultured MB cells. Furthermore, forced overexpression of NRBP2 in MB cell lines causes a dramatic decrease in cell numbers, increased cell death, impaired cell migration and inhibited cell invasion in vitro. Taken together, our data indicate that downregulation of NRBP2 may be a feature by which MB cells escape growth regulation.

## 1. Introduction

Medulloblastoma (MB) is the most common pediatric malignant brain tumor, located in the cerebellum. In spite of surgery and aggressive therapies such as craniospinal radiation and high-dose cytotoxic chemotherapy, about 30% patients succumb to this disease which contributes to around 10% of childhood cancer deaths [1]. Integrative genomic studies have enabled molecular stratification of MB into four subclasses with different prognosis [2,3] and different cells of origin [4]. Expression profiling of MB has shown overlapping gene signatures between cancer cells and stem cells, indicating similar gene regulatory networks in medulloblastoma and neural stem- and progenitor cells (NSPCs) [5].

We previously reported that malignant brain tumors and NSPCs share a common transcriptional signature, and selected the pseudokinase nuclear receptor binding protein 2 (NRBP2) for further study because of its high level of regulation during differentiation of NSPCs [6]. Expression of NRBP2 increases during mouse brain development, and in the cerebellum, NRBP2 is detected in the lining of the fourth ventricle, in a subset of Math1-positive precursor cells around birth, and in the Purkinje cell layer around postnatal day 10. Mature Purkinje cells in the adult mouse cerebellum are intensely stained for NRBP2 [7], and NRBP2 expression thus corresponds to neuronal maturation in the cerebellum. 

Being a pseudokinase, NRBP2 lacks 7 out of 15 residues of the kinase domain [8]. Although devoid of catalytic activity, pseudokinases are increasingly viewed as components of signaling pathways [9]. NRBP2 has a 59% amino acid sequence similarity to nuclear receptor binding protein 1 (NRBP1) a pseudokinase identified as a tumor suppressor and needed for terminal differentiation of intestinal progenitor cells [10]. In addition, NRBP2 has recently been shown to increase the sensitivity of hepatocellular carcinoma cells against chemotherapy, thus suggesting a role in cancer [11].

Because *NRPB2* is a gene under strong regulation during cerebellar differentiation [7], we hypothesized that it could be involved in MB development or progression. Here, we report that there is very little NRBP2 expression in a cohort of brain tumor patients, including MB, and through data-base mining we found that *NRBP2* expression is lower in MB than in the normal cerebellum. Treatment with inhibitors of DNA methylation or histone deacetylation, or RNA knockdown of related factors, revealed that NRBP2 expression is regulated by chromatin-modifying factors in MB. Furthermore, overexpression of NRBP2 increased apoptosis, impaired cell migration and attenuated cell invasion in vitro. Taken together our data indicate that downregulation of NRBP2 is a feature of MB contributing to tumor fitness.

## 2. Results

### 2.1. Low Level of NRBP2 Expression in Human Brain Tumors

Because NRBP2 expression in the mouse brain was higher in differentiated neurons and lower in stem cells, we hypothesized that it might also be expressed at low levels in brain tumor cells, since cancer cells share many properties with NSPCs. Therefore, we performed immunohistochemical staining of a brain tumor tissue array (TMA) with an antibody to NRBP2, followed by annotation by an experienced neuropathologist. The patient cohort contained tumor tissue from 109 patients with 31 different types of brain tumors, including MB (Appendix A). The fraction of stained cells was graded either as 0–1%, 2–10%, 11–25%, 26–50%, 50–75% or >76%. For staining intensity, tumor cores were annotated as negative, weak, moderate or strong. Furthermore, NRBP2 expression was evaluated in the cytoplasmic and nuclear compartment separately. Figure 1A shows that in a majority of the brain tumor tissues, (89 out of 109) less than 1% of the cells are positive for NRBP2 (cytoplasmic staining). Among the remaining 24 samples, only 2 tumor cores exhibit more than 50% stained cytoplasm (Figure 1A, Appendix A). NRBP2 expression was even more rare in the nucleus. Except for three tumors, all tissue cores showed less than 1% NRBP2 staining of the nuclear area (Figure 1A, Appendix A). In the remaining three samples, less than 50% of the total nuclear area was stained for NRBP2. Not only was the NRBP2 stained area minor, the intensity of the NRBP2 staining was mostly graded as weak (Figure 1B, Appendix A), and no sample was evaluated as strong in either cytoplasmic or nuclear compartments. The bubble plot in Figure 1C combines the quantifications from 1A and 1B, to enable comparison of staining quantity (% positive staining) and intensity, of NRBP2 in the nucleus (left) or cytoplasm (right) across all samples. Based on the above findings we conclude that brain tumors express very little NRBP2. In Figure 1D, examples of the low NRBP2 staining in TMA cores are shown; three non-tumor brain cores and six cases of MB illustrate the weak expression of NRBP2. Furthermore, we assessed NRBP2 protein expression by western blot in a set of MB cell lines, and find it to be lower compared to fetal cerebellum (Figure 1E). This was corroborated in a published dataset of RNAseq of the same cell lines, where normal human astrocytes were used as controls (Figure 1F).

### 2.2. Cytoplasmic Localization of NRBP2 in Brain Tumor Cell Lines

To further clarify the cellular localization of NRBP2 protein in brain tumor cells, we used two MB cell lines (D283, D324), a cell line from a central nervous system (CNS) embryonal tumor (PFSK), and a glioblastoma cell culture (U3013MG). For subcellular fractionation, lysates from the cytoplasmic and nuclear compartment were collected separately, and the purity of each fraction was confirmed by expression of ß-tubulin (cytoplasmic) and Lamin A/C (nuclear) (Figure 1G). Consistent with our previous finding in mouse NSPCs [7] and the current brain tumor tissue array, NRBP2 protein, despite its’ name is mostly located in the cytoplasmic compartment.

### 2.3. Reduced NRBP2 Expression in Medulloblastoma Patients Compared to Normal Cerebellum

To explore if low NRBP2 levels is a common trait in MB, we performed data mining, comparing MB and normal tissue. We first used a batch-normalized resource (GSE124814), comprising transcription profiles of 1350 MB cases and 291 normal cerebellum cases [12]. Compared to normal cerebellum, *NRBP2* expression is significantly lower in MB, both when comparing to pediatric cases separately (age < 18 years; *n* = 47), or when combining pediatric and adult cerebelli (all ages; *n* = 291) (Figure 2A). Next, processed gene expression data (GSE85217) that contains a total of 763 MB samples, including 70 WNT cases, 223 SHH cases, 144 Group 3 cases, and 326 Group 4 cases was used [13]. Here, we queried the *NRBP2* mRNA levels within the four MB subgroups, and find significant differences (Figure 2B). Expression of NRBP2 is highest in the wingless (WNT) subgroup, which has the best prognosis, compared to the other three subgroups. Patients from group 4 with intermediate outcome, and group 3 with the worst outcome, express lower levels of NRBP2 than the sonic hedgehog (SHH) subgroup, which has good prognosis in infants and intermediate outcome in other age categories. To further evaluate NRBP2 protein levels in MB, normalized proteomics data (MSV00008264) for a cohort of MB patients was used [14]. Comparison of NRBP2 within the four MB subgroups revealed significant difference between the protein expression in the SHH and G3 subgroup (Figure 2C), similar to the mRNA expression shown in Figure 2B. Taken together, these data show that *NRBP2* expression is downregulated in MB compared to normal brain, and suggests that it correlates to malignancy.

### 2.4. NRBP2 Expression Is Repressed in Medulloblastoma Cell Lines by Epigenetic Factors

Due to the significant down regulation of NRBP2 expression in medulloblastoma patients, we queried publicly available databases for mutations in the *NRBP2* gene that might explain the difference to normal brain. No *NRBP2* mutations were revealed in the three different datasets containing a total of 243 medulloblastoma samples on cBioportal [15,16], and therefore we conclude that *NRBP2* gene alterations in medulloblastoma, if they exist, are rare. On the other hand, epigenetic regulation are universal features of tumors, including MB [17] and large-scale DNA methylation sequencing on human and mouse MB have shown prevalent regions of hypomethylation correlated with increased gene expression, as well as regions displaying increased methylation, causing a decrease in gene expression [18]. We examined if the NRBP2 promoter was under hypermethylation-induced silencing. For this purpose, we used four MB cell lines, one classified as SHH group (D324), two group 3 lines (sD425 and MB002 [19]), and one classified as group 3/4 (D283) [20]. We treated these cell lines with the DNA methylation inhibitor 5′-Aza-2′-deoxycytidine (dAC) and found NRBP2 mRNA expression levels to be significantly upregulated in three out of the four cell lines (Figure 3A), showing that DNA methylation can reduce NRBP2 expression. Using western blot, we note that, in addition to sD425 and D324 cells, for which the mRNA was upregulated by dAC, also in MB002 where there was only a tendency, but no significant increase in NRBP2, that was seen by qPCR, there is an increase in NRBP2 protein (Figure 3C, left panel). Several large-scale screens have reported mutations in histone modification enzymes in human MB-derived cells lines [21,22], and we therefore treated the same cell lines with the histone deacetylase (HDAC) inhibitor valproic acid (VPA). Indeed, HDAC inhibition results in significant upregulation of NRBP2 in all cell lines tested (Figure 3B) and an increase in protein levels were confirmed by western blot for three of these (Figure 3C, right panel). This finding provides evidence that HDAC inhibition can induce a direct or indirect transcriptional repression of NRBP2 in MB cells.

To validate the consequence of dAC- and VPA- mediated NRBP2 upregulation, we monitored the cell numbers in MB002 and sD425 cells, cultured in stem cell medium, after treatment with the two inhibitors (Figure 3D). Addition of VPA leads to a reduced cell proliferation in both lines, and dAC treatment also causes a somewhat slower growth for sD425, but does not significantly alter MB002 growth.

Histone deacetylases form complexes with related co-repressors, nuclear receptor co-repressor 1 (NCOR1) and nuclear receptor co-repressor 2 (NCOR2) (also known as silencing mediator for retinoic acid and thyroid hormone receptors (SMRT) that mediate gene-repression functions [23,24]. Since it has been proposed that repression of NRBP2 expression is mediated by NCOR/SMRT complexes in murine embryonic fibroblasts [25], we studied the effects of siRNA downregulation of individual components of this complex in D324 medulloblastoma cells. Upon transfection with siRNA targeting NCOR and/or SMRT, we first verified the specificity and efficiency of NCOR and SMRT silencing, respectively (Figure 3E, left and middle panels). Next, we found that downregulation of NCOR and the combination of reduced NCOR and SMRT, but not that of inhibiting SMRT alone, release the NRBP2 transcriptional block and increase its mRNA expression (Figure 3E, right panel). Hence, NRBP2 expression seems to be specifically regulated by NCOR in D324 medulloblastoma cells.

### 2.5. Overexpression of NRBP2 Leads to a Reduction in Cell Numbers, Impaired Migration, and Inhibits Invasion In Vitro

Although VPA and to some extent dAC treatment led to reduction in cell growth (Figure 3D) of MB cells, concomitant with an upregulation of NRBP2 expression, it is not possible to define a role of NRBP2 with regard to growth based on the above data since e.g., VPA influences expression of a large number of genes under histone acetylation. Therefore, we used a plasmid to overexpress NRBP2-IRES2-eGFP (plasmid containing NRBP2 in the internal ribosome entry site (IRES; 1, 2) and the enhanced green fluorescent protein (eGFP) coding region) under control of the cytomegalovirus (CMV) promoter in D324 medulloblastoma cells. (Figure 4A). After transfection, the cells were cultured in the presence of selection antibiotics, and to monitor proliferation of NRBP2-overexpressing cells, only cells concomitantly expressing green fluorescent protein (GFP) were counted. Already after the first two days, NRBP2 overexpressing cells tends to increase more slowly than cells transfected with the control vector, and, at day four, the number of NRBP2 overexpressing cells have decreased dramatically (Figure 4B). On the contrary, when NRBP2 was downregulated by a short hairpin RNA (shRNA) (Figure 4C), cell numbers are not affected (Figure 4D).

When we performed a scratch assay to measure the ability of NRBP2-overexpressing D324 cells to migrate, cells were unable to fully cover the empty area during the first 48 h (Figure 4E), and, compared to control cells, NRBP2 overexpressing cells spread more slowly (Figure 4F). Invasiveness is a common denominator of MB, and the ability of cancer cells to break down an artificial extracellular matrix can be used as a proxy for their invasive capacity. D324 cells overexpressing NRBP2 are slower in invading collagen gels than the control cells, as can be seen from Figure 4G,H. Taken together, our data show that forced over-expression of NRBP2 in MB cells reduced cell numbers, as well as their ability to migrate and invade. 

### 2.6. Overexpression of NRBP2 Causes Cell Death in Medulloblastoma Cells

Next, we investigated the reason for reduced cell numbers in NRBP2-overexpressing cells, and found that cell death increases already 24 h post-transfection in D324 cells (Figure 5A). The fraction of dead cells at 48 h is 15% higher in NRBP2-overexpressing cells than in controls. Since NRBP2 overexpressing cells display reduced viability, we asked if the over-expression of NRBP2 had an effect on proliferation. However, there is no difference in the proportions of cells distributed in different cell-cycle stages at 24 or 48 h (Figure 5B). Given that cell death increases at 24 h post-transfection, we also investigated whether NRBP2 over-expression induced proliferation arrest at an even earlier time. Analysis of Ki67 expression at early time-points after transfection (2–24 h) in MB002 and sD425 cells shows unaltered levels of this proliferation marker (Figure 5C). Taken together, we conclude that NRBP2 overexpression did not shift cell proliferation, nor cell cycle parameters. 

Cell viability analysis of NRBP2 overexpressing cells indicates cell death at one to two days post-transfection (Figure 5A). We therefore assessed the activation of AKT, as a general mediator of cell survival, and find AKT phosphorylation to be repressed in NRBP2-overexpressing D324 cells two days post transfection (Figure 5D), further linking the reduced cell numbers to cell death. When we examined the percentage of cells positive for the apoptosis marker Annexin V after NRBP2 overexpression, approximately 25% more positive cells are present on the second day after transfection (Figure 5E). Furthermore, the apoptosis marker cleaved caspase-3 is present in NRBP2-overexpressing cells 48 h post transfection (Figure 5F). We next stained cytospin preparations of MB002 cells with an antibody to cleaved caspase-3, and detect almost no protein before 24 h post transfection (Figure 5G). Furthermore, examination of *BAK1* and *BAX* also showed that these proapoptotic genes increase their expression level upon NRBP2 overexpression as seen by their mRNA (Figure 5H), as well as protein levels (Figure 5I). We therefore conclude that upregulation of NRBP2 induces apoptosis, thereby reducing MB cell numbers. 

## 3. Discussion

It is well accepted that a disturbed NSPC differentiation can cause tumor development [5,26,27,28], and many neural stem cell regulators have subsequently been shown to play important roles in malignant brain tumors [29,30,31]. NRBP2 was found by us to be highly regulated during NSPC differentiation [6], in particular in the cerebellum [7]. Here, we investigated a putative role for NRBP2 in MB because of its expression in cerebellum and its highly regulated developmental expression. We report that NRBP2 is downregulated in MB, and that overexpression in vitro reduces MB cell numbers, and impairs migration and invasion, due to apoptosis.

There is little prior information to place NRBP2 in signaling networks of importance for tumor development, due to the limited published information about this group of pseudokinases [9]. We hypothesized that NRBP2 could be endowed with tumor-reducing capacity, and base this assumption on the findings presented in this study, and a publication on hepatocellular carcinoma [11] that suggested NRBP2 to increase sensitivity to chemotherapy. Publications showing that NRBP1, the other member of this two-protein pseudokinase family, regulates intestinal progenitor cell proliferation, and is downregulated in a variety of human cancers, including colorectal cancers [10,32], lend support to roles for these pseudokinases in inhibiting tumor growth.

NRBP2 staining of brain tumor tissue reveals very low protein expression, and this prompted us to examine published transcriptional profiles of MB versus normal brain [33], where we observe a NRBP2 downregulation that distinguished MB from a non-tumor brain. Because this difference is highly significant, it is possible that downregulation of NRBP2 may reflect an immature, progenitor-like state of MB cells, and that loss, or reduction, of NRBP2 expression during tumor formation relates to an undifferentiated state by which MB aggressiveness increases. Intriguingly, we also find a correlation between lower levels of NRBP2 and molecular subgroups with worse outcomes.

Although it carries a nuclear receptor binding motif, no role for NRBP2 in the nucleus has been reported. Here, we confirm that NRBP2 is mainly localized to the cytoplasmic compartment, which speaks against a role as nuclear receptor-binding activity. The name NRBP2 was given based on its similarity to NRBP1, but neither of these pseudokinases have been shown to bind nuclear receptors.

We presume that downregulation of NRBP2 in MB is mainly due to epigenetic regulation, and base this on the absence of NRBP2 mutations in queried databases, and the lack of reports of NRBP2 in sequencing-based MB studies. In a paper on early postnatal liver development, NRBP2 was one out of three genes where promoter methylation increased in the early postnatal period [34], showing that NRBP2 expression can be regulated by epigenetic modifications during normal development. Furthermore, VPA increases NRBP2 mRNA expression in our experimental system. Cancer-related changes of the histone code have been reported across MB, and HDAC inhibitors have been suggested as candidate therapies [35]. It is therefore interesting that Li et al. [25] in their screen for target genes of nardilysin, a H3K4me2-binding protein, noted NRBP2 as a repressed target in mouse embryonic fibroblasts, presumably through the NCOR/SMRT complex. This is in line with our finding that downregulation of NRBP2 by HDACs involves the co-repressor NCOR.

Overexpressing NRBP2 in MB cells shows its capacity to regulate tumor cell responses, such as reducing migration, invasion, and, most strikingly, cell numbers. Because cell cycle analysis shows no major difference between control cells and NRBP2 overexpressing cells, we propose that the decrease in cell number is mainly due to apoptosis. We base this on our data that Annexin V-positive cells increase, while phosphorylation of AKT decreases, and that cleaved caspase-3, *BAX*, and *BAK1* are upregulated in MB cells as a consequence of increased NRBP2. Furthermore, NRBP2 was found by others to inhibit self-renewal capacity, and AKT phosphorylation, in hepatoceullular carcinoma cells [11].

In summary, the ability of NRBP2 to cause disadvantageous consequences for MB tumor cells suggest that it could be worth investigating ways to increase NRBP2 levels to target several features of medulloblastoma.

## 4. Materials and Methods 

### 4.1. Tissue Microarrays and Analysis

A tissue microarray (TMA), containing duplicate tissue cores (1 mm diameter) from 120 different specimens representing normal human brain tissue and a multitude of various brain tumors, was generated and used for immunohistochemistry-based protein profiling as previously described [36]. The human tissue sections and tumors were obtained from patients diagnosed at Uppsala University Hospital, Uppsala, Sweden, in agreement with approval from the Uppsala Research Ethics Committee, number Epn C-2002-577. Information about the cohorts (diagnosis, gender and age of the patient at time of diagnosis) is given in Appendix A. Histopathologic diagnoses and grade were carefully reevaluated by a neuropathologist (T.O.) in order to get representative samples for the TMA, according to the WHO criteria [37]. Immunohistochemistry using a NRBP2 antibody (Sigma-Aldrich, Stockholm, Sweden) was performed as described [38].

### 4.2. Cell Culture

Medulloblastoma cell lines D324 and D283 were cultured in Dulbecco’s Modified Eagle Medium with 10% fetal bovine serum (FBS). The medulloblastoma cell line MB002 and sD425 (MB004) were kind gifts from Dr. Yoon-Jae Cho (Dept. of Neurology and Neurological Science, Stanford University, USA) and were cultured in 1:1 mixture of Neurobasal without vitamin A (Life Technologies, Stockholm, Sweden) and DMEM/F12 (Life Technologies) supplemented with 1% non-essential amino acids (Life Technologies), 1 mM sodium pyruvate (Life Technologies), 250 mM Hepes (Life Technologies), 1% glutaMAX, B27 (Life Technologies), heparin (Stemcell Technologies, Grenoble, France), leukemia inhibitory factor (Merck Millipore, Billerica, MA, USA), 20 ng/mL of fibroblast growth factor 2 (FGF2) and 20 ng/mL epidermal growth factor (EGF). The neuroectodermal tumor cell line PFSK was grown in RPMI-1640 medium supplemented with 10% FBS and the glioblastoma cell line U3013, part of the HGCC (human glioblastoma cell culture) resource (hgcc.se) was were cultured under neural stem cell conditions as described [39].

### 4.3. Subcellular Fractionation

Cells were first lysed in ice-cold lysis buffer (10 mM morpholineethanesulfonic acid pH 6.2, 10 mM NaCl, 1.5 mM MgCl2, 1mM EDTA, 5 mM dithiothreitol, 1% Triton X-100) with proteinase inhibitor, for 30 min on ice, and then centrifuged for 5 min at 500× *g*. The supernatant containing the cytoplasmic part was collected. The remaining nuclear pellet was first washed once in washing buffer (same as lysis buffer without Triton X-100), then lysed in extraction buffer (25 mM Tris-HCl (pH 10.5), 1 mM EDTA (Ethylenediaminetetraacetic acid), 0.5 M NaCl, 5 mM 2-mercaptoethanol, 0.5% Triton X-100). The cytoplasmic and nuclear fractions were centrifuged at 1500× *g* for 30 min at 4 °C, and the supernatant were stored in −20 °C for further analysis.

### 4.4. Database Mining

In order to compare the gene expression of NRBP2 between normal cerebellum and MB or between MB subgroups, processed gene expression data for three data sets were downloaded from the Gene Expression Omnibus (GEO). The first data set (GSE124814) represents a batch-normalized resource comprising transcription profiles of 1350 MB cases and 291 normal cerebellum cases [12]. The second data set (GSE85217) contains gene expression for a total of 763 MB samples, including 70 WNT cases, 223 SHH cases, 144 Group 3 cases, and 326 Group 4 cases [13]. The third dataset (GSE107405) comprises RNA-seq data on four MB cell lines (D283, D324, MB002 and sD425) in triplicate, as well as single samples of human cerebellar astrocytes (HA-c) and human spinal cord astrocytes (HA-sp) [40]. All statistical analysis was performed on GraphPad software Prism version 6.0 (a commercial proprietary scientific 2D graphing and statistics software, San Diego, CA, USA). For comparison between two groups, a student’s unpaired *t*-test was used. Comparison among all groups was performed with one-way ANOVA, and paired two-sided Welch *t*-tests were used for establishing the significance of differences of the mean gene expression between individual subgroups. 

### 4.5. Treatment with Epigenetic Modifier Drugs

Valproic acid (VPA) and 5-Aza-2′-deoxycytidine (dAC) were from Sigma-Aldrich. VPA was dissolved in 70% ethanol and added to cell cultures daily at a final concentration of 1 mM, with dissolvent as control. dAC was dissolved in DMSO (Dimethyl sulfoxide) and added to cell cultures daily at a final concentration of 1 µM with DMSO as a control. Cell lysates were collected after 48 h.

### 4.6. siRNA Transfection

siNCOR and siSMRT were purchased from Dharmacon (GE Healthcare, Uppsala, Sweden). The D324 cell line was transfected using the protocol suggested by Dharmacon. Cell lysates were harvested after 48 h.

### 4.7. RT-qPCR

RNA was extracted using the RNeasy kit from Qiagen. 500 ng of RNA was used for cDNA synthesis using iScript cDNA synthesis kit (Bio-Rad, Hercules, CA, USA). Quantitative PCR was performed using SYBR Green master mix (Applied Biosystems, Foster City, CA, USA) on a StepOnePlus real time PCR system (Applied Biosystems). Samples were amplified in triplicate and data analyzed using the ΔΔC_T_ method. Primers are listed in Table 1.

### 4.8. Plasmid Transfection

A plasmid for overexpressions of NRBP2 with a V5 tag, under the CMV and linked with IRES-eGFP was produced by Genecopoeia (Rockville, MD, USA). Transfections were performed utilizing lipofectamine 2000 (Life Technology, Stockholm, Sweden). The cell culture medium was changed 24–48 h post-transfection and antibiotics were added to the culture.

### 4.9. Lentivirus Transduction

For targeted downregulation of NRBP2, the sequence CCGGCCCTAAGGACTCATGAGATTACTCGAGTAATCTCATGAGTCCTTAGGGTTTTTTG was used. The double-stranded nucleotide was cloned into a pBMN vector controlled by the H1 promoter and GFP expression driven by the CMV promoter (pBMN-shNRBP2: CMV-GFP-luc-H1-sh NRBP2, pBMN-NTS: CMV-GFP-luc-H1-sh NTS). Lentiviral particles were produced as described [41]. Lentivirus mixed with polybrene (6 μg/mL; Sigma-Aldrich) was added to the cells (70% confluent), and 24 h later, the same procedure was repeated to increase the transduction efficiency of the cell cultures after which cells were transferred to medium with puromycin (1.5 μg/mL; Sigma-Aldrich). NRBP2 overexpressing lentivirus particles were used for overexpressing NRBP2 in MB002 and sD425 cell lines, using the same protocol as described above except that the second transduction round was omitted. For the overexpression virus, the NRBP2 sequence with V5 tag from the Genecopoeia plasmid (described above) was subcloned into a pGH125 vector controlled by EF1A promoter (pGH125-EF1A-NRBP2(V5)-Blastidicin resistance).

### 4.10. Cell Growth Assay

For plasmid transfected cells, cell culture medium was changed and antibiotics were added to the culture 24–48 h after addition of plasmid. The number of GFP positive transfected cells was counted by the Tali image-based cytometer (Life Technologies, Carlsbad, CA, USA), at seeding and 1, 2, and 4 days after medium change. For lentivirus-transfected cells, the serum concentration was decreased to 1% on the second day after the cells were seeded. The cell culture medium was changed every second day. The cell number was counted on day 0 (the second day after cell seeding) 1, 2, and 5.

### 4.11. Scratch Assay

D324 cells transfected with NRBP2 overexpression vectors, and control cells transfected with an empty vector were used in a scratch assay as previously described [38].

### 4.12. Invasion Assay

D324 cells transfected with NRBP2 overexpression vectors, and control cells transfected with an empty vector were used in a collagen invasion assay as previously described [38].

### 4.13. Flow Cytometry

Cells were trypsinized and washed in cold PBS. For cell cycle analysis, cells were fixed in 4% PFA (pH 7.4) for 10 min on ice, then permeabilized in 70% ethanol in 4 °C overnight, and then stained for DNA content by FxCycle staining (Life Technologies, Carlsbad, CA, USA) in the presence of RNase (Life Technologies). For apoptosis analysis, cells were resuspended in binding buffer (10 mM HEPES, 140 mM NaCl, and 2.5 mM CaCl_2_, pH 7.4) and stained with Annexin V (Life Technologies). Cell suspensions were collected on an LSRII cytometer (BD Biosciences, San Jose, CA, US) and analyzed using Flowing Software version 2.5. 1–2 × 10^5^ events were recorded for each analysis.

### 4.14. Western Blot

Cells samples were lysed in RIPA buffer (150 mM NaCl, 1% NP-40, 0.5% deoxycholate, 0.1% SDS, 50 mM Tris, pH7.5) with protease inhibitors (Roche, Basel, Switzerland) and 1 mM Na_3_VO_4_. Protein concentration of cell lysate was estimated by BCA (Thermo Fisher, Bedford, MA, USA). Cell lysates were separated on SDS-PAGE Bis-Tris gels (Life Technologies) and transferred to nitrocellulose membranes (Life Technologies). The membrane was blocked in 5% non-fat milk in TBST (Tris buffered saline supplemented with Tween-20)for 1 h at room temperature and then incubated with primary antibody rabbit anti-NRBP2 (1:1000, Proteintech, Manchester, UK), p-AKT pSer473 (1:000, Cell Signaling Technology, Danvers, MA, USA), cleaved caspase 3 (1:1000, Cell Signaling, MA, USA), mouse monoclonal BAK1 (1:500, Santa Cruz Biotechnology, Dallas, TX, USA), mouse monoclonal Bax (1:500, Santa Cruz Biotechnology), β-Actin (1:500, Santa Cruz Biotechnology), mouse monoclonal Ki67 (1:400, Santa Cruz Biotechnology) overnight at 4 °C. The membrane was washed in TBST and further incubated with peroxidase-conjugated secondary antibody (anti-mouse 1:10,000, anti-rabbit 1:1000, GE Healthcare). The peroxidase activity was detected using either the Amersham ECL Western blotting detection kit (GE Healthcare) or the SuperSignal West Femto Maximum Sensitivity Substrate kit (Thermofisher Scientific, Waltham, MA, USA).

### 4.15. Cell Cytospin and Immunofluorescence 

Cells were collected at various time points and washed in cold PBS, fixed for 10 min in 4% PFA (paraformaldehyde), loaded onto super frost glass slides with the help of a cytofunnel and centrifuged at 600 RPM for 10 min. Cells were allowed to air dry at room temperature and then permeabilized with 0.3% Triton-X for 15 min followed by washing with PBS-T and then 1 h blocking in 5% NGS (normal goat serum) solution before overnight incubation in appropriate antibodies. Subsequently, the slides were washed in PBST and incubated with Alexa Fluor secondary antibodies. The cells were then mounted with Fluoromount-G with DAPI and visualized under the microscope.

## 5. Conclusions

The pseudokinase Nuclear Receptor Binding Protein 2 (NRBP2) was previously identified by us in a screen for genes regulated during neural differentiation. Here we report that NRPB2 expression is low, or absent, in brain tumors, including medulloblastoma (MB). NRBP2 is expressed at a lower level in MB than in the normal cerebellum, and the most malignant MB subtypes have the lowest expression. Overexpression of NRBP2 in MB cell lines caused cell death by apoptosis, and we suggest that downregulation of NRBP2 could be a way for MB cells to escape growth regulation.

## Figures and Tables

**Figure 1 cancers-12-01483-f001:**
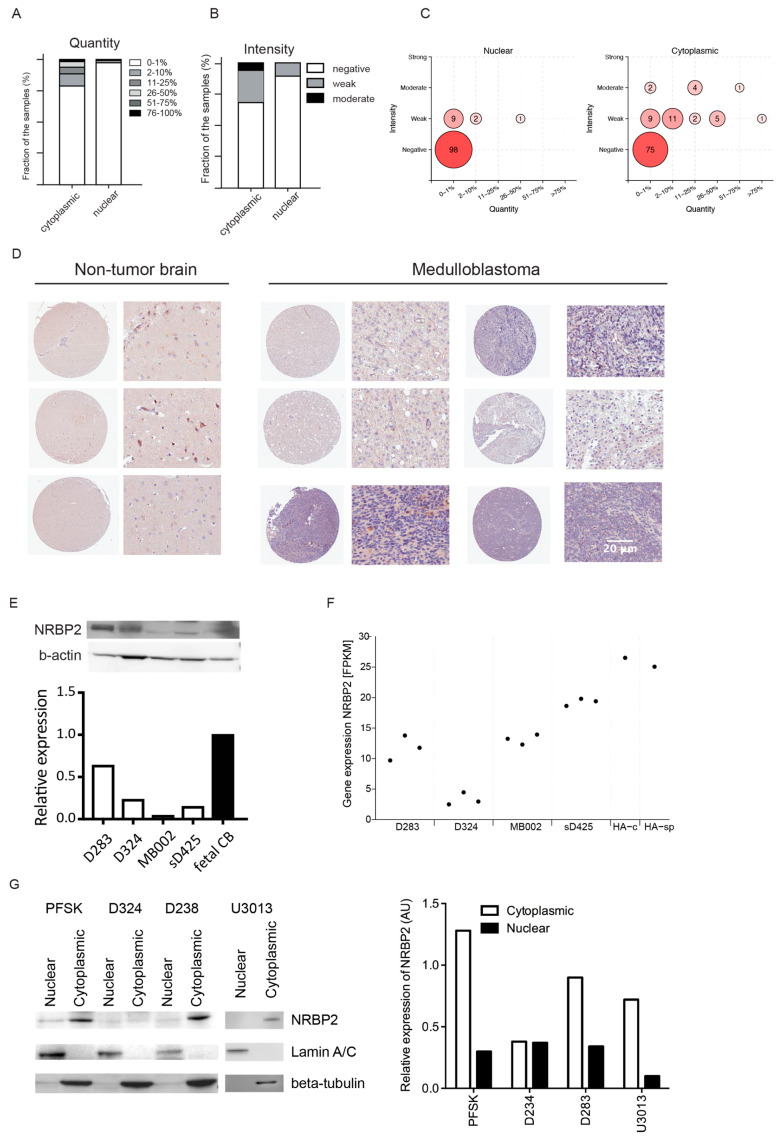
Expression of nuclear receptor binding protein 2 (NRBP2) in human brain tumors. (**A**) Quantification of the NRBP2 positive staining area in human brain tumors; (**B**) Quantification of NRBP2 staining intensity in human brain tumors; (**C**) Bubble plot comparing the staining quantity (% positive staining) and intensity of NRBP2 in the nucleus (left) or cytoplasm (right) across all samples; (**D**) Immunohistochemistry staining of NRBP2 in non-tumor brain and medulloblastoma tissues of the same tissue microarray, scale bar 20 µm; (**E**) Top panel: NRBP2 protein expression (western blot) in fetal human cerebellum and in medulloblastoma cells D283, D324, MB002 and sD425, Bottom panel: Relative NRBP2 protein expression, normalized to b-actin as loading control of the western blot above, setting fetal human cerebellum (CB) = 1; (**F**) *NRBP2* gene expression levels (GSE107405) in MB cell lines (D283, D324, MB002, sD425) in triplicates, and single samples of human cerebellar astrocytes (HA-c) and human spinal cord astrocytes (HA-sp); (**G**) NRBP2 is mainly present in the cytoplasm as seen by subcellular fractionation followed by western blot of cell lysates from a central nervous system(CNS) embryonal tumor cell line PFSK, MB cell lines D283 and D324, and glioblastoma cells U3013MG (Left panel) and the corresponding quantification of intensity (Right panel).

**Figure 2 cancers-12-01483-f002:**
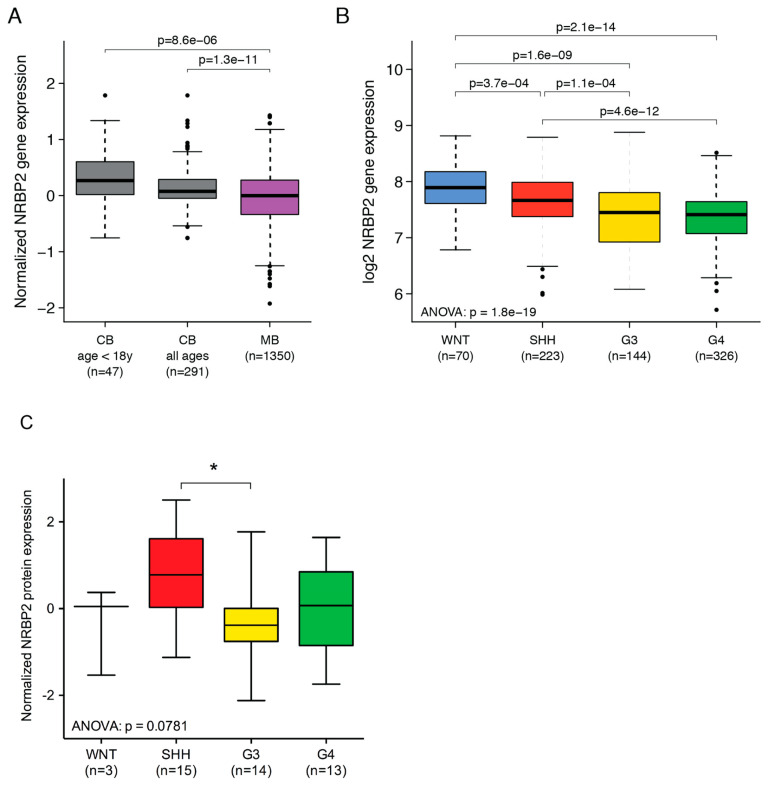
Expression of nuclear receptor binding protein (NRBP2) in human medulloblastoma. (**A**) *NRBP2* gene expression levels (GSE124814) in normal human cerebellum, either for pediatric cases only (age < 18 years; *n* = 47) or across pediatric and adult cases (all ages; *n* = 291), and MB patients (*n* = 1350); (**B**) *NRBP2* gene expression levels (GSE85217) in wingless (WNT) (*n* = 70), sonic hedgehog (SHH) (*n* = 223), Group 3 (*n* = 144), and Group 4 (*n* = 326) subgroups of medulloblastoma patients. Boxes indicate the range between the first and third quartiles, black horizontal lines represent the median value, and whiskers extend to the extreme values excluding outliers, which are shown as dots; (**C**) NRBP2 protein expression levels (MSV00008264) in WNT (*n* = 3), SHH (*n* = 15), Group 3 (*n* = 14) and Group 4 (*n* = 13) subgroups of MB patients. Boxes indicate the range between 1–99 percentile, black horizontal lines represent the median value, and whiskers extend to the extreme values. Unpaired t test between the SHH and G3 groups showed a significant difference (* denotes significance as *p* < 0.05).

**Figure 3 cancers-12-01483-f003:**
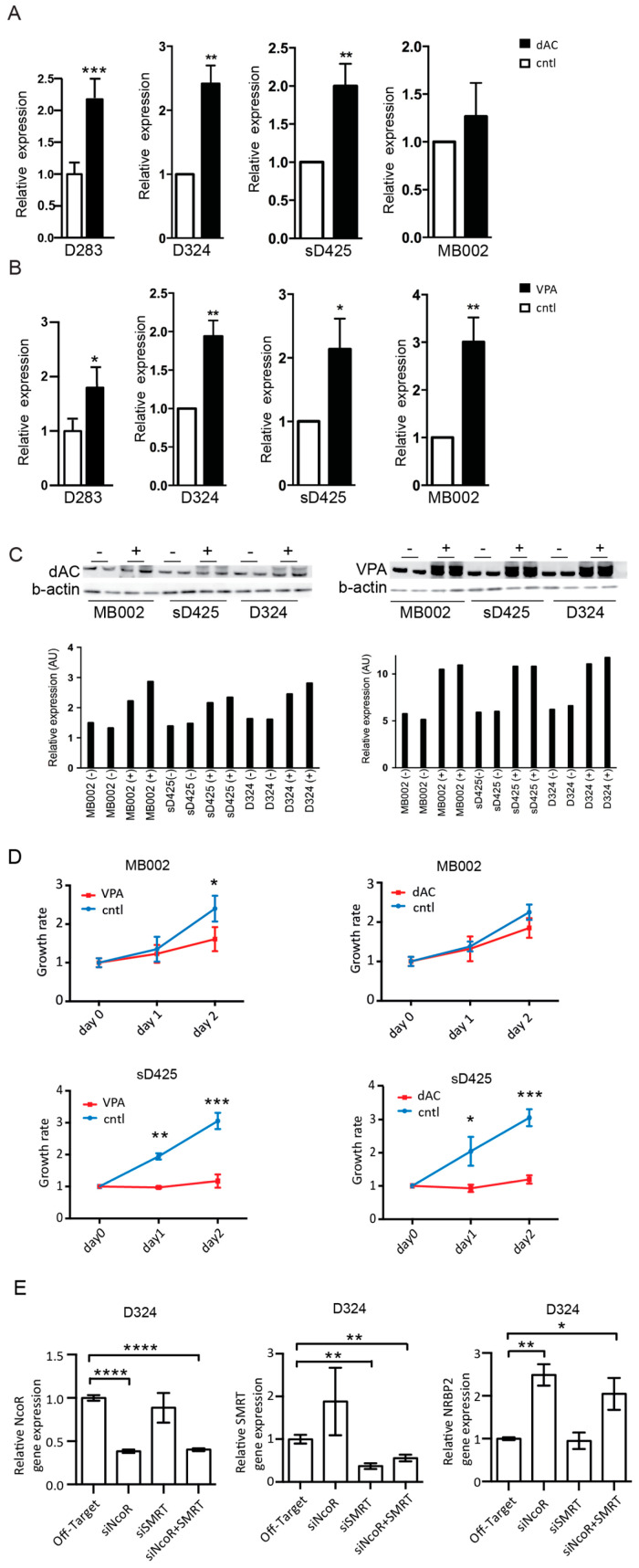
Effects on nuclear receptor binding protein (NRBP2) mRNA expression and cell growth of medulloblastoma (MB) cell lines by promotor de-methylation and histone de-acetylation treatment. (**A**) relative NRBP2 expression in D283, D324, sD425, and MB002 cell lines treated with 5-Aza-2′-deoxycytidine (dAC) for two days, expressed as RQ value, 2-(ΔΔCt) value, beta-actin was used as an endogenous control and untreated cells as the reference. *n* = 3; (**B**) relative NRBP2 expression in D283, D324, sD425, and MB002 cell lines treated with Valproic acid (VPA) for two days, expressed as RQ (relative quantification) value, 2-(ΔΔCt) value, beta-actin was used as an endogenous control and untreated cells as the reference. *n* = 3; (**C**) Left Upper panel: NRBP2 protein expression in MB002, sD425, and D324 cell lines treated with dAC for two days. Left Lower panel: Signal intensity quantification by ImageJ. Right Upper panel: NRBP2 protein expression in MB002, sD425 and D324 cell lines treated with VPA for two days. Right Lower panel: Signal intensity quantification by ImageJ (an open source image processing program developed at the National Institute of Health, USA)**.** b-actin was used as a loading control for both; (**D**) Left panel: growth curves for MB002 (upper) and sD425 (lower) cell lines treated with VPA compared to untreated cells. Right panel: growth curves for MB002 (upper) and sD425 (lower) cell lines treated with dAC compared to untreated cells. * *p* < 0.05, ** *p* < 0.01, *** *p* < 0.001, **** *p* < 0.0001; (**E**) Left panel: relative gene expression levels of NCOR after transfection by off-target-siRNA, NCOR-siRNA, SMRT-siRNA, and NCOR and SMRT-siRNA; middle panel: relative gene expression levels of SMRT after treatment by off-target-siRNA, NCOR-siRNA, SMRT-siRNA, and NCOR and SMRT-siRNA; right panel: relative gene expression levels of NRBP2 after treatment by off-target-siRNA, NCOR-siRNA, SMRT-siRNA, and NCOR and SMRT-siRNA. *n* = 3. Average expression levels after off-target-siRNA transfection were used as control.

**Figure 4 cancers-12-01483-f004:**
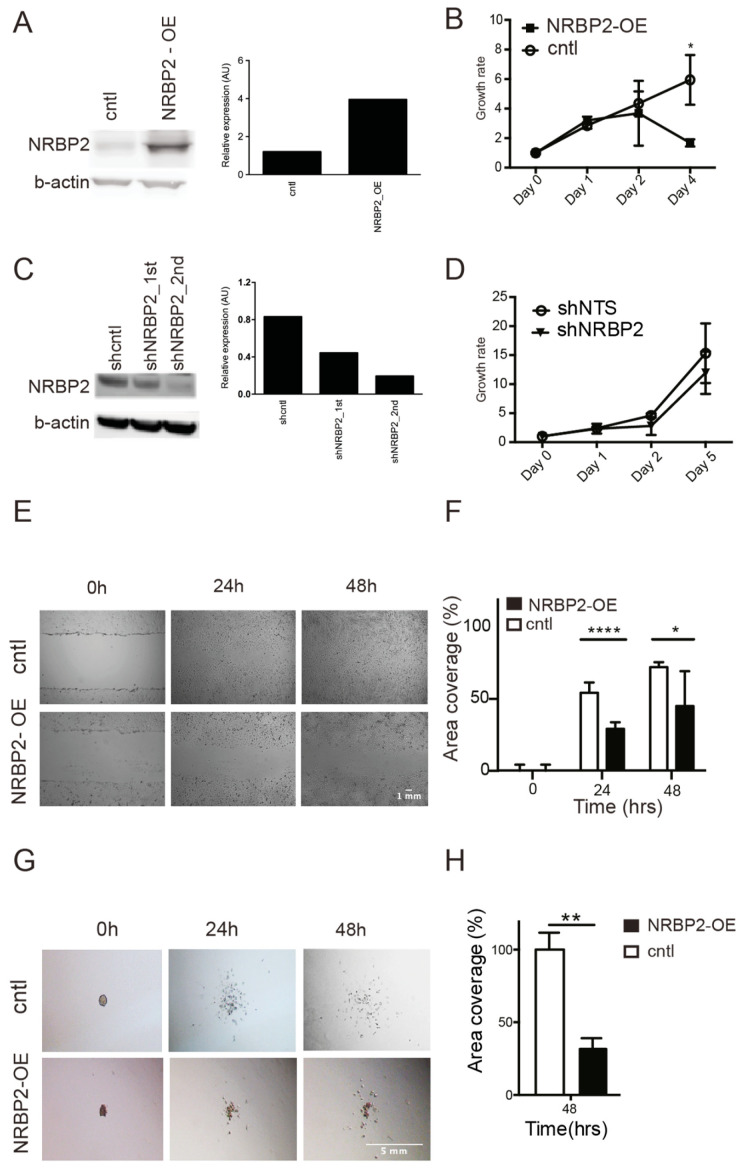
Overexpression of nuclear receptor binding protein (NRBP2) in D324 medulloblastoma (MB) cells causes reduction of cell numbers, migration, and invasion. (**A**) Western blot of NRBP2 in D324 cells transfected with an empty vector (cntl) or NRBP2-IRES2-eGFP NRBP2 sequence with V5 tag and linked with the internal ribosome entry site (IRES2) and the enhanced green fluorescent protein (eGFP) coding region) overexpression plasmid, Left panel: Signal intensity quantification by ImageJ; (**B**) growth rate of D324 cells transfected with control vector or NRBP2 vector; (**C**) Western blot of NRBP2 in D324 transduced with scrambled shRNA, or shRNA against NRBP2. The cultures were transduced twice to increase the efficiency of downregulation, Left panel: Signal intensity quantification by ImageJ; (**D**) growth curve of D324 cells transduced with scrambled shRNA, or shRNA against NRBP2; (**E**) photomicrographs of cell migration after scratching the cell monolayer (*n* = 3); (**F**) quantification of coverage of cell-free area from (**C**) (*n* = 3); (**G**) photomicrographs of cell invasion in collagen gel at different time points of D324 cells transduced with scrambled shRNA, or shRNA against NRBP2; (**H**) quantification of invasion from (**E**) (*n* = 3). * *p* < 0.05, ** *p* < 0.01, **** *p* < 0.0001.

**Figure 5 cancers-12-01483-f005:**
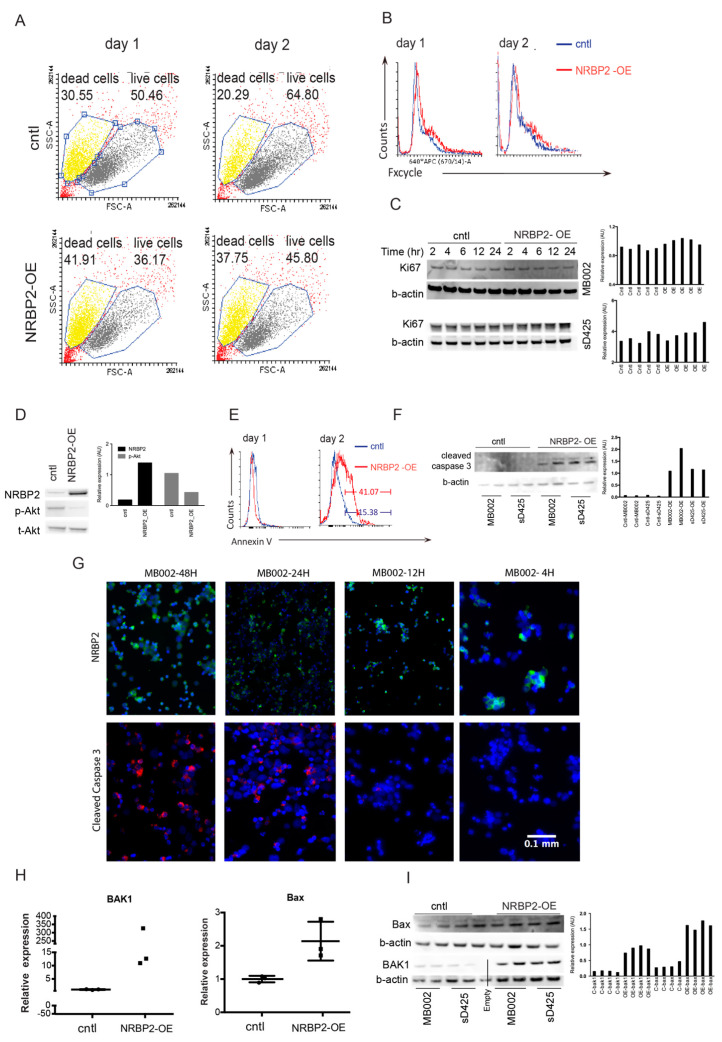
Overexpression of nuclear receptor binding protein (NRBP2) in medulloblastoma (MB) leads to increased cell death. (**A**) flow-cytometry-based quantification of live and dead D324 cells 1 and 2 days post-transfection with NRBP2 -V5-IRES-eGFP (NRBP2 sequence with V5 tag, linked with the internal ribosome entry site (IRES2) and the enhanced green fluorescent protein (eGFP) coding region) plasmid; (**B**) flow-cytometry-based cell-cycle analysis in D324 cells after transfection with NRBP2 -V5-IRES-eGFP plasmid; (**C**) Western blot of Ki67 in MB002 and sD425 cells 2–24 h post transduction with empty vector or EF1A-NRBP2(V5)-plasmid. Corresponding Signal intensity quantification by ImageJ on the left; (**D**) Western blot of phosphorylation of AKT in D324 cells transfected with control or NRBP2 plasmid and signal intensity quantification by ImageJ on the left; (**E**) apoptosis analysis based on Annexin V labelling of control-transfected or NRBP2-transfected D324 cells; (**F**) Western blot of cleaved caspase-3 in MB002 and sD425 cells transduced with empty vector or NRBP2 overexpressing plasmid, 48 h post transduction; (**G**) cytospin preparations of MB002 cells, 4 to 48 h after transduction with a EF1A-NRBP2(V5)- plasmid showing staining for NRBP2 (upper panel) and cleaved caspase-3 (lower panel); (**H**) expression of apoptotic genes *BAK1* and *BAX* mRNA in D324 cells transfected with control or NRBP2 plasmid; (**I**) Western blot of BAK1 and BAX in MB002 and sD425 cells transduced with control or NRBP2 plasmid 48 h post transduction. Corresponding signal intensity quantification by ImageJ is shown on the left.

**Table 1 cancers-12-01483-t001:** List of sequences of the primers used for qPCR.

NRBP2	F 5′-GGCCTCATCAAGATCGGCTC-3′
	R 5′-GCAGGTTCCGAAGTTCCTCTC-3′
NcoR	F 5′-GGAAGACTACCATTACTGCAGCTAACT-3′
	R 5′-CATCCTTGTCCGAGGCAATT-3′
SMRT	F 5′-GGGTAAATATGACCAGTGGGAAGAG-3′
	R 3′-TGGCATTCAGAGGGTTAAAAGC-3′
BAX	F 5′-CCCGAGAGGTCTTTTTCCGAG-3′
	R 5′-CCAGCCCATGATGGTTCTGAT-3′
BAK1	F 5′-GTTTTCCGCAGCTACGTTTTT-3′
	R 5′-GCAGAGGTAAGGTGACCATCTC-3′
b-actin	F 5′-CTAAGGCCAACCGTGAAAAG-3′
	R 5′-ACCAGAGGCATACAGGGACA-3′

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
