# Peer review of "Nuclear Receptor Binding Protein 2 Is Downregulated in Medulloblastoma, and Reduces Tumor Cell Survival upon Overexpression"

_cancers, 2020, doi:10.3390/cancers12061483_

Round 1

Reviewer 1 Report

The authors added some new data in the revised manuscript. Although these new data improve the manuscript to some degree, the major concerns are not addressed. For example: Major concern No. 1. "Based on only six MB cases, the authors draw a conclusion that the level of NRBP2 is decreased in human compared to normal cerebellum. The authors should significantly increase the MB case numbers." In the revised manuscript , there are still six MB cases.

Moreover, there is no convincing evidence showing that the level of NRBP2 is decreased in human MB. The authors should use increased number of human MB samples, and then perform reliable quantitative assay such as western blot and real-time PCR to demonstrate the level of NRBP2 is decreased in human MB.

Additionally, Figure 1E are very rough and unconvincing.

Author Response

Dear Editors,

We are grateful for the invitation to revise our manuscript: Nuclear Receptor Binding Protein 2 is down-regulated in Medulloblastoma, and reduces tumor cell survival upon overexpression, by A. Xiong et al, and below we outline the point-by-point response to the reviewer’s comments. Based on the comments by Reviewer 1, we have revised the manuscript, performed additional analysis, and added new figures. From the online review report, we are happy to note that Reviewer 2 is satisfied with our revision of the manuscript. We hope that the manuscript is now suitable for publication.

Sincerely,

Karin Forsberg-Nilsson

REVIEWER 1

The authors added some new data in the revised manuscript. Although these new data improve the manuscript to some degree, the major concerns are not addressed. For example: Major concern No. "Based on only six MB cases, the authors draw a conclusion that the level of NRBP2 is decreased in human compared to normal cerebellum. The authors should significantly increase the MB case numbers." In the revised manuscript, there are still six MB cases.

Moreover, there is no convincing evidence showing that the level of NRBP2 is decreased in human MB. The authors should use increased number of human MB samples, and then perform reliable quantitative assay such as western blot and real-time PCR to demonstrate the level of NRBP2 is decreased in human MB.

We fully agree with Reviewer 1 that it was necessary to increase the number of MB analyzed for NRBP2 expression. Therefore, we had included expression data based on 1350 MB cases and 291 normal cerebelli in the revised manuscript, compared to previously 117 MB cases and normal cerebellum from five individuals, i.e. almost ten times more. We believe that this considerably strengthens our data and that we convincingly show that NRBP2 mRNA expression is significantly lower in MB than in the normal cerebellum.

In addition, we have now analyzed published proteomic data for NRBP2 expression of 45 MB patients included in Archer et al, Cancer Cell 2018 (Sep 10;34(3):396-410.e8. doi: 10.1016/j.ccell.2018.08.004). This reveals that NRBP2 protein levels have similar inter-subtype expression relationships as the NRBP2 mRNA, with the exception of WNT MB where the study only had three cases of this subtype. These data are now included as a new Fig. 2C.

Another source of protein expression is The Human Protein Atlas (www.proteinatlas.org), a comprehensive resource containing normal and pathological tissues. The pathology atlas contains adult glioma, but unfortunately no pediatric brain tumors. Similar to our data (Fig. 1A-C),

glioma tissues included in HPA express low NRBP2 levels,

https://www.proteinatlas.org/ENSG00000185189-NRBP2/pathology, while the normal human brain has medium expression of NRBP2 protein

https://www.proteinatlas.org/ENSG00000185189-NRBP2/tissue.

  1. Additionally, Figure 1E are very rough and unconvincing.

In support of the western blot for NRBP2 expression in MB cell lines and normal brain (Fig 1E), we have now added RNA-seq data for the same cell lines (new Fig 1F). This shows NRBP2 mRNA expression in MB cell lines D283, D324, MB002, and sD425, in triplicates, compared to single samples of human cerebellar astrocytes (HA-c) and human spinal cord astrocytes (HA-sp). The data was retrieved from a larger RNA-seq dataset (GSE107405) published in Bolin et al, Oncogene 2018 (PMID: 29511348) and confirms lower NRBP2 expression in the MB cell lines, compared to normal controls.    

Reviewer 2 Report

n/a

Author Response

Dear Editors,

We are grateful for the invitation to revise our manuscript: Nuclear Receptor Binding Protein 2 is down-regulated in Medulloblastoma, and reduces tumor cell survival upon overexpression, by A. Xiong et al, and below we outline the point-by-point response to the reviewer’s comments. Based on the comments by Reviewer 1, we have revised the manuscript, performed additional analysis, and added new figures. From the online review report, we are happy to note that Reviewer 2 is satisfied with our revision of the manuscript. We hope that the manuscript is now suitable for publication.

Sincerely,

Karin Forsberg-Nilsson

REVIEWER 1

The authors added some new data in the revised manuscript. Although these new data improve the manuscript to some degree, the major concerns are not addressed. For example: Major concern No. "Based on only six MB cases, the authors draw a conclusion that the level of NRBP2 is decreased in human compared to normal cerebellum. The authors should significantly increase the MB case numbers." In the revised manuscript, there are still six MB cases.

Moreover, there is no convincing evidence showing that the level of NRBP2 is decreased in human MB. The authors should use increased number of human MB samples, and then perform reliable quantitative assay such as western blot and real-time PCR to demonstrate the level of NRBP2 is decreased in human MB.

We fully agree with Reviewer 1 that it was necessary to increase the number of MB analyzed for NRBP2 expression. Therefore, we had included expression data based on 1350 MB cases and 291 normal cerebelli in the revised manuscript, compared to previously 117 MB cases and normal cerebellum from five individuals, i.e. almost ten times more. We believe that this considerably strengthens our data and that we convincingly show that NRBP2 mRNA expression is significantly lower in MB than in the normal cerebellum.

In addition, we have now analyzed published proteomic data for NRBP2 expression of 45 MB patients included in Archer et al, Cancer Cell 2018 (Sep 10;34(3):396-410.e8. doi: 10.1016/j.ccell.2018.08.004). This reveals that NRBP2 protein levels have similar inter-subtype expression relationships as the NRBP2 mRNA, with the exception of WNT MB where the study only had three cases of this subtype. These data are now included as a new Fig. 2C.

Another source of protein expression is The Human Protein Atlas (www.proteinatlas.org), a comprehensive resource containing normal and pathological tissues. The pathology atlas contains adult glioma, but unfortunately no pediatric brain tumors. Similar to our data (Fig. 1A-C),

glioma tissues included in HPA express low NRBP2 levels,

https://www.proteinatlas.org/ENSG00000185189-NRBP2/pathology, while the normal human brain has medium expression of NRBP2 protein

https://www.proteinatlas.org/ENSG00000185189-NRBP2/tissue.

  1. Additionally, Figure 1E are very rough and unconvincing.

In support of the western blot for NRBP2 expression in MB cell lines and normal brain (Fig 1E), we have now added RNA-seq data for the same cell lines (new Fig 1F). This shows NRBP2 mRNA expression in MB cell lines D283, D324, MB002, and sD425, in triplicates, compared to single samples of human cerebellar astrocytes (HA-c) and human spinal cord astrocytes (HA-sp). The data was retrieved from a larger RNA-seq dataset (GSE107405) published in Bolin et al, Oncogene 2018 (PMID: 29511348) and confirms lower NRBP2 expression in the MB cell lines, compared to normal controls.  

This manuscript is a resubmission of an earlier submission. The following is a list of the peer review reports and author responses from that submission.

Round 1

Reviewer 1 Report

The authors investigate the pseudokinase NRBP2 in medulloblastoma and describe findings that suggest there is an inverse correlation between expression levels and fitness of medulloblastoma tumor cells.

This is a very well written paper of an interesting topic. The scientific method is sound and the conclusions are supported by the findings. 

On page 2, line 68 dollar signs ($) have be inadvertently placed in the subtitle.

Reviewer 2 Report

Xiong et al. investigate the role of NRBP2 in medulloblastoma (MB). They show that the level of NRBP2 is decreased in MB compared to normal cerebellum. They also find that treatment of MB cell lines with drugs inhibiting DNA methylation histone deacetylation increase the expression of NRBP2. Moreover, they find that enforced xpression of NRBP2 in MB cell lines leads to a dramatic decrease in cell numbers, increases cell death, impaires cell migration, and inhibites cell invasion in vitro. Based on the findings, they conclude that down-regulation of NRBP2 is a feature by which pediatric tumors escape growth regulation. Although the topic is potential interesting, the manuscript has a number of major problems.

Based on only six MB cases, the authors draw a conclusion that the level of NRBP2 is decreased in human compared to normal cerebellum. The authors should significantly increase the MB case numbers. The authors determine cytoplasmic Localization of NRBP2 in MB cells using MB Cell Lines. The authors should use primary MB cells. Figure 2, although data mining of published dataset is useful and importantly, the authors should perform follow-up experiments to verify their findings. The authors should perform real-time PCR and/or western blot to verify the RNA sequencing data from published dataset. The authors show that enforced expression of NRBP2 in MB cell lines leads to a dramatic decrease in cell numbers, increased cell death, impaired cell migration and inhibited cell invasion in vitro. Does enforced expression of NRBP2 in MB cell lines affect their viability and invasion in vivo. Why the DNA methylation inhibitor 5'-Aza-2'-129 deoxycytidine (dAC) influences NRBP2 mRNA expression in two out of the four cell lines? Does enforced expression on NRBP2 in MB cell affect cell proliferation? Figure 4 G should be removed and replaced.

Minor problem

There are a number of typos, such as “There is Low Leve$l of NRBP2 expre$ssion in hum$an bra$in tu$mors”.

Reviewer 3 Report

In the study entitled “Nuclear Receptor Binding Protein 2 is Down-2 Regulated in Medulloblastoma, and Reduces Tumor 3 Cell Survival upon Overexpression”, Xiong et al. studied nuclear receptor binding protein 2 (NRBP2) is mostly expressed in normal cells in cerebellum, but not in tumor cells in medulloblastoma (MB). The authors further showed inhibition of DNA methylation and histone decetylation by the inhibitors resulted in upregulating NRBP expression. Furthermore, forcedly expressed NRBP2 in MB cell lines led to increased cell death and inhibiting cell migration and invasion. This study could provide a new epigenetic target for testing DNA methylation and HDAC inhibitors in the treatment of MB.

The major issues include:

 1) There is no in vivo experiment to test whether overexpression of NRBP2 in MB cells can suppress xenograft tumor growth, metastasis, and prolonging animal survival.

 2) In vivo testing of DNA methylation and HDAC inhibitors is required for checking whether tumor cells with low level of NRBP2 are sensitive to these drug treatments.

 3) The authors showed that overexpression of NRBP2 induced cell death, however, it is confused whether attenuation of migration and invasion in NRBP2 overexpressed cells are because of cell death, particularly in 24 and 48 hours. The authors should examine short period of time, e.g. 2-6 hours.

4) The authors should show promoter analysis of NRBP2 for hypermethylation, and check histone modifications of NRBP2 in MB cell lines and normal cells.

The other minor issues include:

  • There are multiple “$” in line 68.
  • The quality of images in Fig 1C needs to be improved. The quantification and statistics for NRBP2 staining in normal and tumors tissues are required (e.g. IHC sore etc.). In addition, more MB tissues should be stained (only 6 MB being used in this study (e.g. MB TMA).
  • The multiple public datasets should be used for subgroup analysis in Fig 2.
  • Patient outcome should be analyzed based on NRBP2 mRNA expression by using these datasets.
  • Western blotting analysis should be used for testing NRBP2 expression in Fig 3A and B.
  • The additional shRNA should be used for confirming NRBP2 knocking down in Fig 4G.
  • Regarding overexpression of NRBP2 induced apoptosis, PI and annexin v should be used together to check early apoptosis and late cell death in Fig 5D.
  • Besides Bak and Bax, the authors should other classic markers, such as caspase 3, BAK, BAX in WB as well in Fig 5E.